# Solar Position Algorithm Based on the Kepler Equation

**Weidong Huang *** and **Bowen Liu**

Anhui Province Key Laboratory of Polar Environment and Global Change, Department of Environmental Science and Engineering, University of Science and Technology of China, 96 Jinzhai Road, Hefei 230026, China; liubw@mail.ustc.edu.cn

\* Correspondence: huangwd@ustc.edu.cn; Tel.: +86-551-63606631; Fax: +86-551-63607386

**Abstract:** When calculating the position of the sun, earth's motion can be assumed to be an ellipse if the accuracy of calculation is required to be 0.01 degrees. Then, Kepler's equation can be applied from the mean anomaly of the sun at a specific time to calculate the true anatomy of the sun at the time, and the sun's position can be calculated. The average absolute error of calculating the sun's altitude and azimuth is only 0.04 and 0.06 mrad, respectively, which can meet the requirements of a concentrated solar tracking system. This method only needs to correct the length of the regression year and the near point year for every 100 years, so it can be used for a long time.

**Keywords:** sun position; solar tracking; azimuth

## 1. Introduction

Accurate calculation of the position of the sun plays an important role in solar energy applications, especially for a concentrated system, which can be directly used for tracking system control [1]. The accuracy required by various applications is very different. A non-concentrated system can tolerate several degrees of error without causing obvious loss, while a concentrated system needs an error of 0.01°(0.175 mrad). A few important applications, such as the calibration of some basic data with the observation instruments, require higher accuracy [2].

Although it is simple on the surface, it is a rather difficult task to accurately calculate the position of the sun [3]. In fact, the apparent motion of the sun is affected by a lot of disturbances, such as the precession and nutation of the earth's axis of rotation, disturbances caused by the moon, a decrease in the earth's rotational speed, and the influence of other planets. All these factors affect the calculation in different ways. Considering all these factors, the calculation becomes very complicated, and it is difficult to apply in solar engineering. Usually, on the premise of meeting the engineering requirements, some approximations are often applied to simplify the calculation.

In the literature of solar energy engineering, algorithms with different precision and complexity can be found to calculate the position of the sun [4]. These calculation methods can be divided into two categories.

The first type of method is to directly calculate the solar declination and equation of time with an empirical formula [5–10] and then calculate the sun position of the observation point after it is converted to a ground coordinate system. In the first type of method, the formulae proposed by Spencer in 1971 [7] are the most widely cited in the solar literature. The solar declination calculated by the Spencer algorithm shows that the maximum error can be 0.28°. The error can be reduced by using a complex empirical formula. Based on the equations of *The Astronomical Ephemeris* and *The American Ephemeris and Nautical Almanac* (1961), Pitman and Vant-Hull [10] presented an algorithm with an accuracy of 0.02°. Recently, a very simple model for the position of the sun was provided by Shapiro [11] with an error of up to 2.15° in elevation and up to 3.1° in the azimuth, which may be acceptable for calculating the absorption by sunlight collectors.

The second method is to calculate the solar longitude and latitude with an empirical formula, convert it to a celestial coordinate system to calculate the solar right ascension and declination, and then convert it to a ground coordinate system to calculate the position of the sun. For example, the PSA algorithm [12] given by Blanco et al. was determined ti have an accuracy of 0.5′ over the period of 1999–2015. It has been updated to achieve a 25% decrease in the average error of the angular deviation with respect to the true solar vector (from a mean error of 11.81–8.78 arcsec) [13]. The SPA program [14] developed by the American Renewable Laboratory has one of only $0°.0003$. This has become the standard high-accuracy implementation used by the solar community. SG2 is a new algorithm for the sun position with an accuracy c. $0.0015°$ from 1980 to 2030, provided by Blanc and Wald [15]. This is achieved by devising approximations of the original equations of the SPA to decrease the number of operations. However, the SPA and SG2 are both too complex and slow for those needing continuous high-accuracy solar tracking using small processors, such as processors driving heliostats. Grena [16] gave five new algorithms with different accuracies from $0.19°$ to $0.0027°$ over the period from 2010 to 2110. They can consider many disturbing factors, such as nutation, aberration, and so on, and the calculation accuracy can reach a very high level. Hoadley recently developed the SG2, SPA, and Grena's G5 algorithm with rectangular coordinates, but it reduced the computational complexity a little [3].

The common disadvantages of these two kinds of methods are that their calculation methods are all empirical, the accuracy is greatly affected by the calculation year, and the time range used is limited. Once it exceeds the applicable range, the error may become too large to be applied.

The method proposed in this paper is an analytical method based on Kepler's equation, which can be easily calibrated and obtain enough precise results that it can be applied for a long time. This paper summarizes the main calculation formulas of the method, evaluates the calculation errors, and estimates the calculation complexity of the method.

## 2. Calculation Principle and Method

The basic principle is that when the accuracy of calculation is required to be 0.01 degrees, for calculating the position of the sun, it can be assumed that the motion of the earth is a pure ellipse; that is to say, the perturbation of the moon and planets is ignored. According to Kepler's equation and the sun's near point angle at a specific time, the near point angle and the true near point angle at any time can be calculated such that the solar ecliptic can be calculated. Then, the right ascension and declination can be calculated by coordinate conversion to a celestial coordinate system and then converted to a ground coordinate system to calculate the altitude and azimuth of the sun at the observer's position [17]. The earth's rotation is regarded as uniform, and its non-uniformity and long-term slowdown are ignored; that is to say, the equation of gravity is used to calculate the inhomogeneity of the earth's revolution. In addition, the empirical equation is used to calculate the influence of nutation and aberration. In this paper, two methods are described, and the difference lies in whether the influence of the nutation of the earth is considered or not. Within 200 years, the calculated annual average sun position error was evaluated to be less than 0.1 mrad.

*Calculation Process*

First, the sun position data of the initial point are selected. In this paper, taking 12:00 p.m. GMT on 1 January 2000 as the initial point, the geometric mean longitude of the sun is calculated according to the initial value and the time interval calculated by the length of the tropical year:

$$L0 = 280°.46645 + 360° * JC \tag{1}$$

$$JC = (JD - 2451545.0)/365.24218968 \tag{2}$$

Among these equations, the following is true:

$$JD = INT(365.25(Y + 4716)) + INT(30.6001(M + 1)) + D + 2 - A + INT(A/4) - 1524.5 \tag{3}$$

$$A = INT(Y/100) \tag{4}$$

where INT stands for the rounding of omitted decimals and D includes the specific time data, with the day as unit.

We can then calculate the mean anomaly of the sun according to the initial value and the time interval from the initial point:

$$m = 357.52910 + 360.0 * (JD - 2451545.0)/365.25963586 \tag{5}$$

The eccentricity of earth's orbit is calculated as follows:

$$e = 0.016708617 - 0.0000420388 * T \tag{6}$$

The sun is situated in the focus S of the elliptical orbit, K is situated at K, and C is the center of the ellipse, as shown in Figure 1. Therefore, Kepler's equation is as follows.

$$E = m + e * sin(E) \tag{7}$$

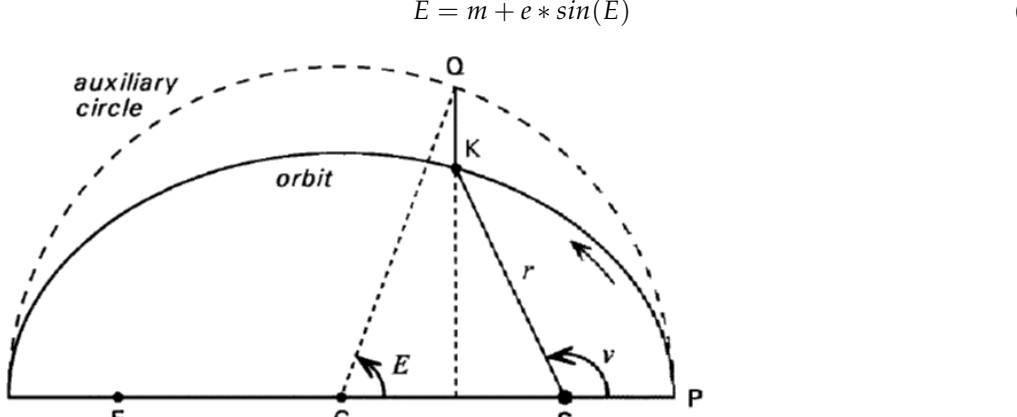

**Figure 1.** The eccentric anomaly E and the true anatomy v definition for Kepler's equation [17].

As the eccentricity $e$ of the earth's elliptical orbit is very small—only about 0.0167—the approximate solution of the eccentric anomaly $E$ can be obtained according to Kepler's equation as follows:

$$tanE = sinm/(cosm - e) \tag{8}$$

The error of the formula is less than 0.2″, which meets the calculation requirements. The true anatomy $v$ is

$$tan(v/2) = [(1+e)/(1-e)]0.5 * tan(E/2) \tag{9}$$

The sun's longitude can be calculated as follows:

$$\Theta = L0 + v - m \tag{10}$$

In order to obtain the apparent longitude $\lambda$ of the sun in the ecliptic coordinates, we should also correct the nutation and aberration of $\Theta$:

$$\lambda = \Theta - 0°.00569 - 0°.00478 * sin(\Omega) \tag{11}$$

The latter nutation revision in the above formula increases the calculation amount, and $\Omega$ is the longitude of the ascending node of the moon's mean orbit on the ecliptic, which is measured from the mean equinox of the date:

$$\Omega = 125°.04452 - 19°.34177 * JC \tag{12}$$

In the simplified version of this method, it is ignored in calculation. Only the aberration is considered, so the above formula is simplified as



$$\lambda = \Theta - 0°.00569 \tag{13}$$

The latitude of the sun in the ecliptic coordinates does not exceed $1''.2$, which can be set to 0. Therefore, the solar right ascension $\alpha$ and declination $\delta$ can be calculated by coordinate transformation using the following formula:

$$tan\alpha = cos\varepsilon sin\lambda / cos\lambda \tag{14}$$

$$sin\delta = sin\varepsilon sin\lambda \tag{15}$$

where $\varepsilon$ is the obliquity of the ecliptic or inclination of the earth's axis of rotation and is the angle between the equator and the ecliptic, which is calculated as follows:

$$\varepsilon = 23°26'21''.448 - 0''.46816 * T + 9''.20cos(\Omega) + 0''.57cos(2L0) \tag{16}$$

The latter two items are for nutation correction, which can also be ignored in further approximations. In addition, the quadrant of the angle $\alpha$ should be carefully calculated, and $\alpha$ can be directly calculated by using the ATN2 function in a computer language; that is, $\alpha$ = ATN2(cos($\varepsilon$)*sin($\lambda$), cos($\lambda$)).

The hour angle $\omega$ is calculated as follows:

$$\omega = L0 - \alpha - 0°.0057183 + (hour - 12) * 15° + (l - utc) * 15° - 17''.2 * sin(\Omega) - 1''.32sin(2L0) \tag{17}$$

where the last two terms are negligible nutation correction, l is the longitude of the observation point, *hour* is the time of the calculation moment, and the unit is hours. Finally, we convert this to a ground coordinate system to calculate the solar altitude h and azimuth A:

$$Sinh = sin\delta sin\phi + cos\delta cos\varphi cos\omega \tag{18}$$

$$sinA = cos\delta sin\varphi / cosh \tag{19}$$

where $\varphi$ is the latitude of the observation point.

The final step is the computation of the refraction correction. It is independent of the main body of the algorithm, and the formula in [14], presented here, can be replaced with any other formulas that better describe the local conditions or specific meteorological situations. The formula given here depends on the local atmosphere pressure $P$ and temperature $T$:

$$\Delta h = \frac{P}{1010} \frac{283}{273 + T} \frac{\frac{1'.02}{60}}{tan[h + \frac{10.3}{h+5.11°}]} \tag{20}$$

where $P$ is the local pressure (in mbars), $T$ is the local temperature (in °C), and $h$ and $\Delta h$ are all in degrees.

### 3. Error Evaluation

The SPA program developed by the American Renewable Laboratory was used to evaluate the error of this method. The error of the SPA program is only $0°.0003$, which is about two orders of magnitude lower than that of this method. Compared with this method, its error can be neglected, and its application time range is from $-2000$ to 4000 years [14], which can be used to evaluate the accuracy of this method.

The SPA program needs input parameters such as the geographical location and time and can obtain solar position parameters such as the solar altitude and azimuth. The average error and standard deviation of the algorithm were obtained by comparison with the SPA proposed by NREL.

### 4. Calculation Results

As shown in Figure 2, this was used to calculate the absolute error distribution of the results obtained at different times in 2020 with the SPA method. For every day, 160 times

were calculated from 7:12 a.m. to 4:48 p.m. solar time every 3.6 min. The latitude of the calculated position was 37°.1, and the longitude was −2°.36. The average absolute error of the solar altitude angle was 0.031 mrad, and the average absolute error of the azimuth angle was 0.042 mrad. If the nutation effects were ignored, the two errors would increase to 0.037 and 0.048 mrad, respectively.

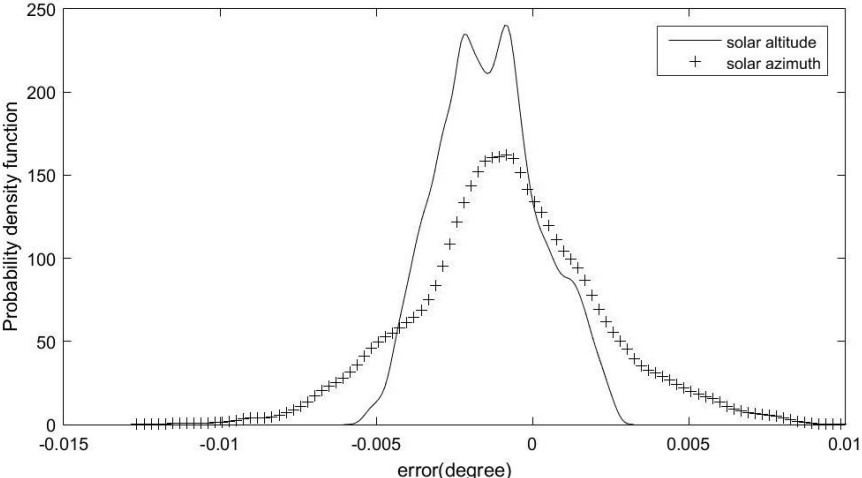

**Figure 2.** The probability density distribution of the calculation error of the solar altitude and azimuth in 2020, where 160 times were calculated from 7:12 a.m. to 4:48 p.m. solar time every 3.6 min. Latitude = 37°.1, and longitude = −2°.36.

Figures 3 and 4 show the average absolute error and standard deviation of the calculation of the solar altitude and azimuth from 2010 to 2210, respectively. As can be seen from the figure, the average error for the solar altitude was about 0.04 mrad, while for the azimuth it was about 0.06 mrad. The standard deviation of the calculation results of the sun's azimuth was larger than the altitude's, and the error distribution range of the azimuth was larger than that of the altitude.

According to the calculation scale provided by Grena [4], the computational complexity of this method is 765, and when ignoring nutation correction, the computational complexity is 701.9.

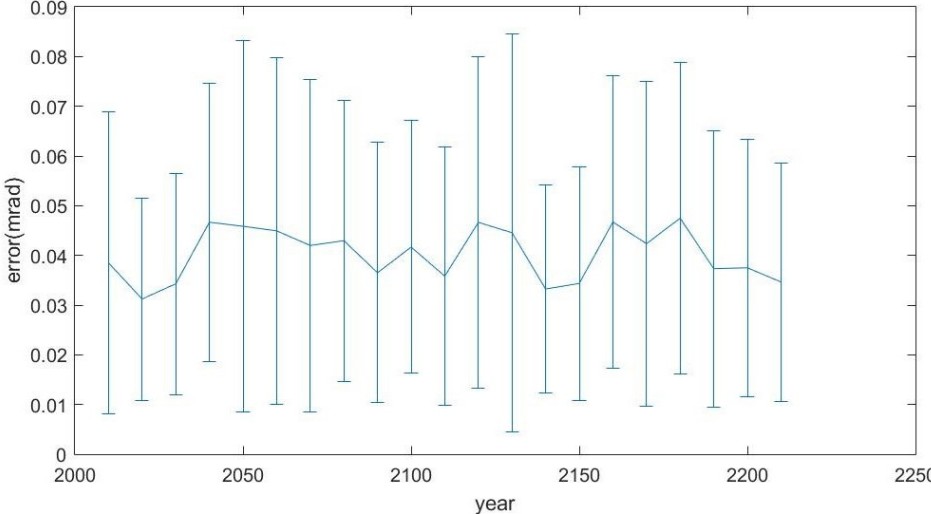

**Figure 3.** Average absolute error and standard deviation of solar height calculation from 2010 to 2210. Latitude = 37°.1, and longitude = −2°.36.

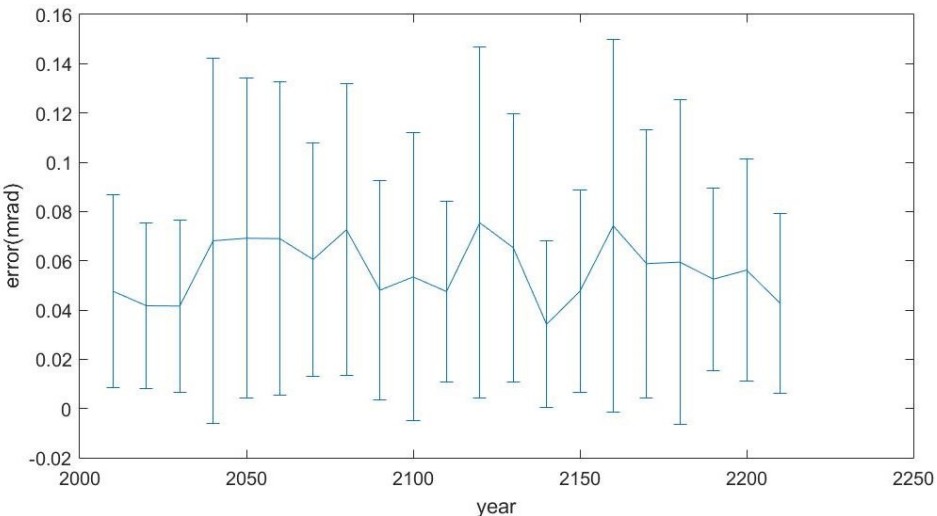

**Figure 4.** Average absolute error and standard deviation of solar azimuth calculation from 2010 to 2210. Latitude = 37°.1, and longitude = −2°.36.

## 5. Discussion

The optical error of the solar concentrating optical system was about 1–6 mrad. Even if the minimum optical error of 1 mrad and the maximum tracking error of 0.20 mrad were calculated, the optical error of the system would increase from 1 mrad to 1.0248 mrad, only increasing by 2.48%. The tracking error calculated by the average absolute error only increased by 0.132%, so the error caused by the tracking system could be ignored. Therefore, this method meets the tracking requirements of the current concentrated solar optical system.

As shown in Table 1, when compared with the five calculation methods provided by Grena [4], the error of this method was only lower than that of the fifth method, and it was superior to the other four methods. The computational complexity of this method was lower than that of the fifth method and higher than those of the other four methods.

**Table 1.** Comparison of performance of different algorithms in determining the sun's position

|  | **Grena's Method 3** | **Grena's Method 4** | **Grena's Method 5** | **Prewent Method Based on Kepler** |
|---|---|---|---|---|
| Error in zenith Average (degree) Range (degree) | 0.0021 [−0.0113, 0.0119] | 0.0015 [−0.0091, 0.0093] | 0.0005 [−0.0025, 0.0027] | 0.0023 [0, 0.005] |
| Error in azimuth Average (degree) Range (degree) | 0.0043 | 0.0038 | 0.0005 | 0.0034 [−0.0006, 0.0092] |
| Time validity (year) | 2010–2110 | 2010–2110 | 2010–2110 | 2010–2510 |
| Computational cost | 572 | 641 | 929 | 765 |

Due to the non-uniformity of the earth's rotation and revolution, we may need to regularly calibrate the length of the regression year and the near-point year for about 100 years. According to the data provided by the astronomical almanac, we can complete the calibration, which can meet the accuracy requirements once for every 100 years. In addition, the empirical formula of the earth's eccentricity and the intersection angle between the equator and the ecliptic is used in this method, and the calculation errors of the two formulas within 1000 years are kept below 0.1 mrad, so it can be applied for more years.

## 6. Conclusions

This method is an analytical method obtained by solving Kepler's equation. Although several key data need to be calibrated regularly due to the irregularity of the earth's rotation, these data are easily obtained from observation data or an astronomical almanac, and it is not necessary to reconstruct the method. Moreover, the time interval required for calibration is longer than the traditional empirical method. This algorithm has a similar accuracy to the PSA algorithm, but it can work for more than 500 years. It can be said that it is a more universal calculation method. It has less than 0.1 mrad of error for computing the position of the sun, which can meet the tracking requirement of the concentrated solar system.

**Author Contributions:** Conceptualization, W.H.; methodology, W.H.; software, W.H.; validation, W.H. and B.L.; investigation, W.H.; resources, W.H.; data curation, W.H.; writing—original draft preparation, W.H.; writing—review and editing, W.H. and B.L. All authors have read and agreed to the published version of the manuscript.

**Funding:** This work is partially supported by the National Natural Science Foundation of China (No. 11574292). The part of numerical calculations in this paper has been done on the supercomputing system in the Supercomputing Center of University of Science and Technology of China.

**Institutional Review Board Statement:** Not applicable.

**Data Availability Statement:** Not applicable.

**Conflicts of Interest:** The authors declare no conflict of interest.

## Nomenclature

| | |
|---|---|
| A | Azimuth |
| e | Eccentricity of the earth's orbit |
| E | The approximate solution of the eccentric anomaly |
| v | True anatomy |
| h | Solar altitude |
| INT | The integer of the calculated terms |
| JC | The Julian century |
| JD | The Julian day |
| L0 | The geometric mean longitude of the sun |
| l | Longitude |
| m | The mean anomaly of the sun |
| P | Local atmospheric pressure (mbars) |
| T | local temperature |
| Y | The year |
| D | The day of the month with decimal time |
| M | The month of the year |
| $\alpha$ | Solar right ascension |
| $\delta$ | Declination |
| $\Delta h$ | The refraction correction |
| $\varepsilon$ | Obliquity of the ecliptic |
| $\Theta$ | Sun's longitude |
| $\lambda$ | The apparent longitude of the sun in the ecliptic coordinates |
| $\varphi$ | Latitude |
| $\omega$ | Hour angle |
| $\Omega$ | The longitude of the ascending node of the moon's mean orbit on the ecliptic |

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
