# Peer review of "Solar Position Algorithm Based on the Kepler Equation"

_applsci, doi:10.3390/app12115449_

Round 1
Reviewer 1 Report
[1]-In the introduction, the authors' references to the review of other research are not sufficient.
[2]-The authors mentioned the use of the SPA program, they need to elaborate more on how it is used to assess the error of this method. What parameters were considered?
[3]-The conclusion of the study is not given.
[4]-In the references, only 6 references, please give more. This is not enough.
Reviewer 2 Report
This paper presents an analytical method based on Kepler’s equation to calculate the position of the sun with an accuracy that meets the requirements of concentrating solar tracking system. This method, thanks to a calibration of the length of the regression year and the near-point year for about 100 years, can be used for a long time.
In general, the manuscript is well organized and written and the research interest is clear and fully understandable. The literature review is relevant to the research even if the introduction could provide further references. The method is well detailed even though its impact should be further discussed: in lieu of the paragraph “4. Discussion and summary”, you could consider to write a paragraph “Discussion” to address better the relevance of this method compared to the others available in the literature (for example showing them in a table with features or pros/cons) and then a paragraph “Conclusion”.
Finally, please report sources for non-calculated numbers indicated in the equations and please report the missing units of measure in the “Nomenclature” (e.g. T: annual average local temperature).

Reviewer 3 Report
- The underlying research problem is not entirely justified in the introduction section. It is not clear why this method is particularly useful to the body of knowledge, given the state of the art.
- It is required to show a graphical representation demonstrating the proposed algorithm or its application; this will help to understand the range of method applicability.
- The pros and cons of the proposed algorithm are simply not compared with any of the well-established state-of-the-art algorithms, making it difficult for the reviewer to evaluate its relevance and usefulness.
- The manuscript figures such as Figure 1 need to be clearly described.
Round 2
Reviewer 3 Report
Thanks for considering the comments